# xView3-SAR: Detecting Dark Fishing Activity Using Synthetic Aperture Radar Imagery

**Fernando S. Paolo**[1,*]**, Tsu-ting Tim Lin**[2,*]**, Ritwik Gupta**[3,4,*,†]**, Bryce Goodman**[3]**,
Nirav Patel**[3]**, Daniel Kuster**[2]**, David Kroodsma**[1]**, Jared Dunnmon**[3]

[1]Global Fishing Watch, [2]Cambrio, [3]Defense Innovation Unit, [4]UC Berkeley

## Abstract

Unsustainable fishing practices worldwide pose a major threat to marine resources and ecosystems. Identifying vessels that do not show up in conventional monitoring systems—known as "dark vessels"—is key to managing and securing the health of marine environments. With the rise of satellite-based synthetic aperture radar (SAR) imaging and modern machine learning (ML), it is now possible to automate detection of dark vessels day or night, under all-weather conditions. SAR images, however, require a domain-specific treatment and are not widely accessible to the ML community. Maritime objects (vessels and offshore infrastructure) are relatively small and sparse, challenging traditional computer vision approaches. We present the largest labeled dataset for training ML models to detect and characterize vessels and ocean structures in SAR imagery. xView3-SAR consists of nearly 1,000 analysis-ready SAR images from the Sentinel-1 mission that are, on average, 29,400-by-24,400 pixels each. The images are annotated using a combination of automated and manual analysis. Co-located bathymetry and wind state rasters accompany every SAR image. We also provide an overview of the xView3 Computer Vision Challenge, an international competition using xView3-SAR for ship detection and characterization at large scale. We release the data (https://iuu.xview.us/) and code (https://github.com/DIUx-xView) to support ongoing development and evaluation of ML approaches for this important application.

## 1 Introduction

Recent advances in remote sensing technology have allowed fishing activity to be tracked across the globe via the Automatic Identification System (AIS) that can broadcast vessels' location [15]. The use of AIS, however, varies by region and fleet; not all vessels are required to carry AIS [37] while some turn off their AIS to engage in illicit activities [30]. This unknown number of non-broadcasting vessels that are not tracked by conventional monitoring systems—referred to as "dark" vessels— limits our ability to effectively manage marine resources. Illegal, Unreported, and Unregulated (IUU) fishing comprises more than 20% of all catch around the world [2]. In recent years, the largest IUU fishing offenses were perpetrated by fleets that mostly did not use AIS [30], costing legitimate fishermen and governments billions of dollars while also damaging critical ecological systems.

Satellite imagery provides an alternative means of sensing dark vessels. Common electro-optical satellites, however, are limited by cloud coverage and low-light conditions. Synthetic Aperture Radar (SAR) satellites, on the other hand, can image in all weather conditions and at night. The European Space Agency (ESA) Sentinel-1 radar satellites cover most coastal waters around the world every 12 days (with a 6-day repeat cycle if combining the two satellites), offering open access to the full

---

*Equal contribution
†Corresponding author, ritwik.ctr@diu.mil

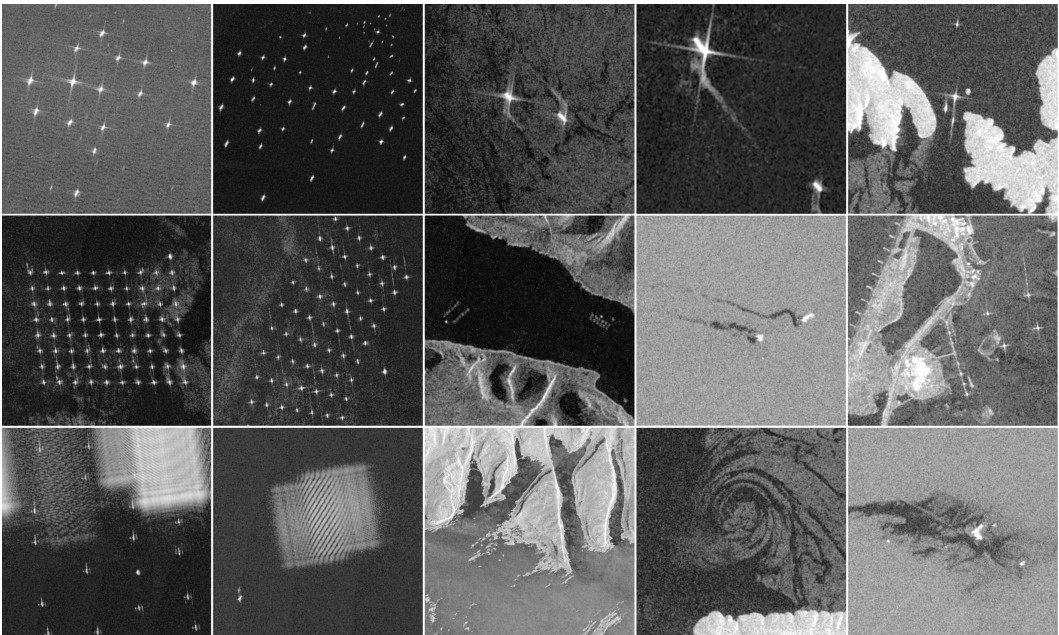

Figure 1: Example of objects and features in xView3-SAR: (top row) vessels of different size, type, and brightness with different backgrounds; (middle row) fixed infrastructure such as wind farms, fish cages, platforms, and port towers; (bottom row) noise artifacts, rough coastlines, rocks, and wind-driven ocean features.

Sentinel-1 SAR archive. Despite its increasing availability, there are still two significant barriers to using SAR imagery at large scale for maritime object detection and characterization with automated computational methods.

The first barrier is generating analysis-ready pixels. SAR actively beams from a moving sensor; the radar waves in turn interact with moving objects on the ground and interfere with themselves, resulting in images that contain characteristic features inherent to the image formation process, such as speckle noise and visible discontinuities. Multi-polarization SAR images can appear notably different from the most commonly used RGB images produced by optical satellites. Processing and interpreting SAR images require domain expertise, as a series of computationally expensive and domain-specific prepossessing steps are needed prior to analysis [40].[1] This limits the access that machine learning (ML) practitioners have to analysis-ready SAR pixels.

The second barrier is ground-truthing maritime objects. Many of these objects are dark vessels not broadcasting their location and, therefore, absent from public records. Vessels are relatively small objects, appearing in medium-resolution SAR images as a few bright pixels scattered through large areas on a cluttered background. For instance, annotated bounding boxes account for only 0.005% of the image pixels in our dataset, which is in stark contrast with what is commonly encountered in computer vision datasets for conventional object detection (e.g., MS-COCO [19]). It is also difficult— if not impossible—to manually annotate for key tasks like length estimation and whether a vessel is a fishing vessel. Thus, practical annotation approaches beyond manual labeling are required to feasibly create large-scale, richly annotated datasets.

Due to these challenges, existing datasets to support the development of computational models for detecting and characterizing dark vessels are extremely limited in size and scope.

We aim to break these barriers by releasing **xView3-SAR**, the largest dataset of its type by orders of magnitude[2] (see Table 7 in Appendix E for a summary of related datasets). We combined (i) global AIS data, (ii) a state-of-the-art AIS-to-SAR matching algorithm, and (iii) expert human-analyst verification to construct xView3-SAR: a multi-modal ship detection and characterization dataset comprised of (1) 991 full-size analysis-ready Sentinel-1 SAR images averaging 29,400-by-24,400

---

[1]For further information on SAR we refer the reader to NASA and ESA resources [12, 27].

[2]xView3-SAR contains 1,400 gigapixels; the next largest SAR vessel detection dataset contains 12 gigapixels.

pixels each, (2) 243,018 verified maritime objects in over 43.2 million square kilometers, (3) detailed annotations for a novel multitask problem formulation that reflects the priorities of anti-IUU fishing practitioners, (4) co-located surface wind condition and bathymetry rasters that provide valuable context for the primary tasks, and (5) a set of reference code to expedite users' development process and dataset extension.

The fact that most IUU fishing vessels do not broadcast their positions, greatly limits the usefulness of labeling approaches relying solely on AIS data for detecting and characterizing dark vessels. We overcome this limitation by taking a hybrid labeling approach that combines an AIS-to-SAR matching algorithm with expert human-analyst verification. This enables ML models trained on the xView3-SAR dataset to "learn" a larger variety of vessel features, including those of dark targets, and detect vessels regardless of their AIS broadcasting status.

Despite the simplicity and efficiency of conventional Constant False Alarm Rate (CFAR) detection algorithms, the standard approach for vessel detection on SAR images [10, 11], there are two main reasons to seek more modern methods. First, it is difficult to implement an automated CFAR algorithm at scale where SAR images often have different statistical properties (see below). Second, any object characterization, such as length and type, must be performed as a secondary task post CFAR detection. ML approaches, in particular neural networks, can learn the statistical properties of the images and automatically determine what constitutes an anomaly. They can also perform regression and classification tasks jointly with object detection, enabling a learned representation to be optimized for all of these tasks simultaneously. For these reasons, ML approaches to vessel detection and characterization in SAR images are highly desirable; this motivates the creation of the xView3-SAR dataset.

In this work, we also provide a high-level overview of the results from the xView3 Computer Vision Challenge, an international competition using the xView3-SAR dataset to detect and characterize dark vessels at large scale.[3] The Challenge brought awareness of the IUU fishing problem to the wider ML community, and resulted in the development of accurate and efficient models that have been deployed in real-world anti-IUU fishing applications. These models provide a useful benchmark for the ML community and shed light on areas for future research; they also highlight the contributions of xView3-SAR in bridging the fields of ML research and remote sensing in the fight against IUU fishing.

## 2   Related Work

Ship detection on satellite imagery is a well-explored problem. A widely used approach for SAR imagery is the Constant False Alarm Rate (CFAR) method [17, 29], which characterizes the statistical properties of the sea clutter to separate the background pixels from the targets of interest. This statistical analysis can be either theoretical, e.g. by defining a probability density function that describes the backscatter properties of the image, or empirical, e.g. by computing the local mean and standard deviation of background pixels [10]. Previous work [9, 20, 46, 52] have also proposed thresholding, shape, and texture-based methods to identify ships in optical and SAR imagery, while [10, 24, 39] have implemented wavelet transforms and spectral analysis to better separate the background backscatter from the foreground, improving target detectability. These conventional approaches, however, require substantial experimentation in order to find optimal setups for a specific data type, usually involving human analysts with domain expertise to evaluate the results. These methods do not adapt nor scale well to new environments or satellite imaging systems, and are not robust to "unseen" data artifacts.

More recently, deep learning methods for computer vision have provided a compelling new approach to the problem of ship detection [3, 25]. In particular, convolutional neural network architectures have been successfully used, with the most common strategies adopting either a single-stage approach where a predefined image grid or anchor boxes are scanned and each cell/box is evaluated for object presence all in one pass [5, 7, 42, 49]; or a dual-stage approach where regions of the image are selected for further analysis and then classified (and retained or excluded) in a secondary pass [21, 50]. Alternatively, [8] combines a modified generative adversarial network with a single-stage detector to produce state-of-the-art results for small-ship detection. [38] further separates the detection problem into two independent tasks, first using a deep neural network to extract features for ships and then passing those features to a downstream model for classification.

---

[3]Hereafter referred to as xView3 for short, with xView3-SAR denoting the dataset.

These approaches have been mostly applied to (and developed for) small-scale problems using a limited number of images, partly because large annotated SAR datasets are difficult to construct. It is particularly hard to annotate SAR images with the level of detail needed for vessel characterization (versus detection) due to the challenge of fusing SAR with AIS information. Approaches to this fusion problem range from correlating AIS pings to ships present in a given area and time [26], to projecting a ship's likely position to the SAR image timestamp by interpolation/extrapolation [23, 31].

Most publicly available datasets for ship detection are based on optical imagery. Examples include the HRSC2016 dataset, which provides 2,976 instances of vessels across 22 classes with detailed segmentation masks [22], and [50], which extends the methodology of HRSC2016 to 5,175 instances of vessels in Google Earth and Bing Maps imagery with oriented bounding boxes. The Airbus Ship Detection dataset [1] provides instance segmentation masks for 192,556 instances of vessels in optical imagery; however, it lacks a classification hierarchy for vessels. Uniquely, the SeaShips dataset [35] contains ground-based optical imagery with 40,007 instances of vessels within six different classes.

Some optical satellite image datasets not tailored specifically for maritime domain awareness also contain instances of maritime objects. For instance, xView1 [16] provides WorldView imagery with 5,141 instances of vessels split across nine classes in its training set, and DOTA v1.0 [45] aggregates imagery from multiple sources to provide 37,028 instances of ships with oriented bounding boxes.

There are also a few datasets composed of SAR imagery tailored specifically for vessel detection. Examples include the LS-SSDD-v1.0 [47], which provides 15 SAR images from Sentinel-1 with 6,015 expert-annotated ship bounding boxes, manually verified with AIS and Google Earth co-incident imagery where available. FUSAR-Ship [13] uses a sophisticated spatio-temporal Hungarian matching scheme to annotate 1,851 instances of vessels on Gaofen-3 SAR imagery with AIS beacon pings. We provide a thorough comparison of related optical and SAR datasets in Table 7 (Appendix E).

## 3  Dataset Construction

We used SAR imagery from the ESA Sentinel-1 mission, which comprises two near-polar-orbiting satellites imaging 24/7 and covering most coastal waters every 12 days (each). SAR imaging systems can function in different "acquisition modes," each with a different primary application and resolution-coverage trade-off. We used the Interferometric Wide (IW) swath mode Level-1 Ground Range Detected (GRD) product, which is freely available through ESA[4] and NASA[5] data portals. This imagery has a spatial resolution of about 20 meters with pixel spacing of 10-by-10 meters. Note that small vessels within single pixels may display high backscatter intensity extending beyond the pixel limit (Figure 1).

---

[4]https://scihub.copernicus.eu
[5]https://search.asf.alaska.edu

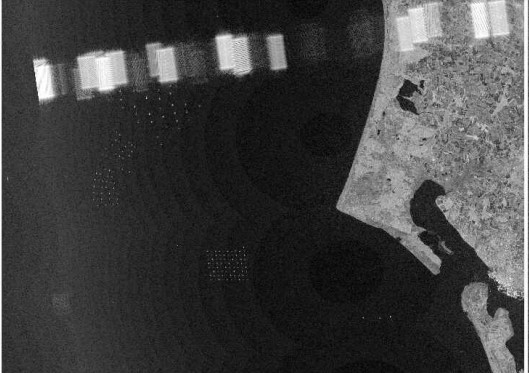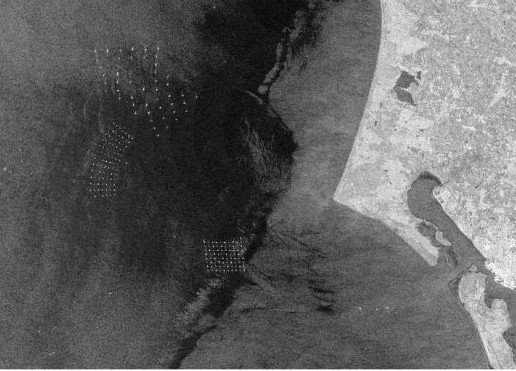

Figure 2: Excerpt of a Sentinel-1 SAR image showing both polarization bands: VH (left) and VV (right). Note how different features appear on each band, e.g., an artifact across the top portion of the VH band that may have been caused by, for example, ground radio-frequency interference, and wind-driven features on the ocean surface are visible in the VV band. Both bands depict clearly the patterns of fixed infrastructure (likely wind farms). Location is near Esbjerg, Denmark.

The IW mode GRD product includes both the vertical-horizontal (VH) and vertical-vertical (VV) polarization bands for each image. The cross-polarization channels (VH or HV bands) represent the proportion of the signal that changes its polarization before it is received due to interacting with objects at the surface. Over relatively flat areas, such as the ocean surface, only a small fraction of the signal returns polarized. Thus, the VH band usually shows a better separation between vessels (strong returns of polarized signal) and the sea clutter (weak polarized signal) making it well-suited for vessel detection. The co-polarization channels (VV or HH bands) are signals vertically or horizontally transmitted and received by the sensor. In this case, small variations in surface texture can produce varying backscatter behaviors, highlighting sea surface features such as oil slicks, sediment plumes, and wind-driven structures, providing useful context for ship characterization. The difference between the VH and VV bands used in xView3-SAR can be observed in Figure 2.

Our workflow for constructing xView3-SAR is as follows: (1) select strategic geographic areas; (2) process the raw imagery; (3) process the ancillary data; (4) detect objects with an automated CFAR algorithm; (5) correlate AIS data to SAR detections; (6) classify AIS data to characterize vessel type and activity; (7) manually label images; (8) combine manual and automated annotations; (9) partition the data for training and validating ML algorithms (Figure 3). Each of these steps are described in detail below.

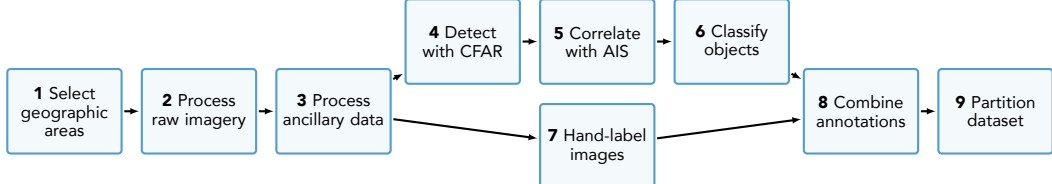

Figure 3: Dataset creation process; see Section 3 for description of each step.

**Step 1. Select geographic areas**. In order to provide a consistent SAR training dataset for ML detection and characterization problems, we need areas with comprehensive SAR and AIS coverage. Subject to these constraints, we selected strategic geographic areas capturing different vessel types, traffic patterns, latitudes, and a variety of fixed infrastructure (Figure 4). We included several areas near Europe because (1) European waters are fully covered with high frequency by Sentinel-1, and (2) many European vessels broadcast their positions through AIS, which serves as an abundant source of high-confidence labels for SAR imagery (see below). These regions include the North Sea, Bay of Biscay, Iceland, and the Adriatic Sea. We also included images from West Africa which are regions with high IUU activity and substantial offshore oil development. In all, the xView3-SAR dataset contains 991 full-size SAR images that are, on average, 29,400-by-22,400 pixels, surpassing the size of most existing object detection datasets. We acknowledge that xView3-SAR does not offer a true global coverage and therefore may not represent all vessel distributions. We attempted to balance, however, data availability and quality with diversity of objects and activities.[6]

**Step 2. Process raw images**. Raw Sentinel-1 images require a series of computationally-intensive and domain-specific processing steps to create ML-ready rasters. We used ESA's Sentinel-1 Toolbox to derive backscatter values (in dB) for each pixel from the original unprojected GRD product. This includes orbit correction, removal of noise artifacts, radiometric calibration, terrain correction, and reprojection to the WGS84 reference system [28]. We provide all images in UTM projection, with 10-meter pixel spacing for the VH and VV rasters, distributed as GeoTIFFs with an original file size of approximately 2.4 GB per band. We used half-precision floating point to facilitate data download, resulting in a 50% reduction in file sizes.

**Step 3. Process ancillary data**. In addition to the VH and VV SAR images, we provide a set of ancillary rasters to aid the ship detection task, and support novel approaches to context-aware analytical models (Figure 11). The original Sentinel-1 Level-2 Ocean (OCN) product, containing surface wind information, is available from ESA at lower resolution with a pixel spacing of 1 km. The bathymetry product obtained from the General Bathymetric Chart of the Oceans [6] consists of

---

[6]Our SAR processing pipeline is available in the xView3 GitHub repository.

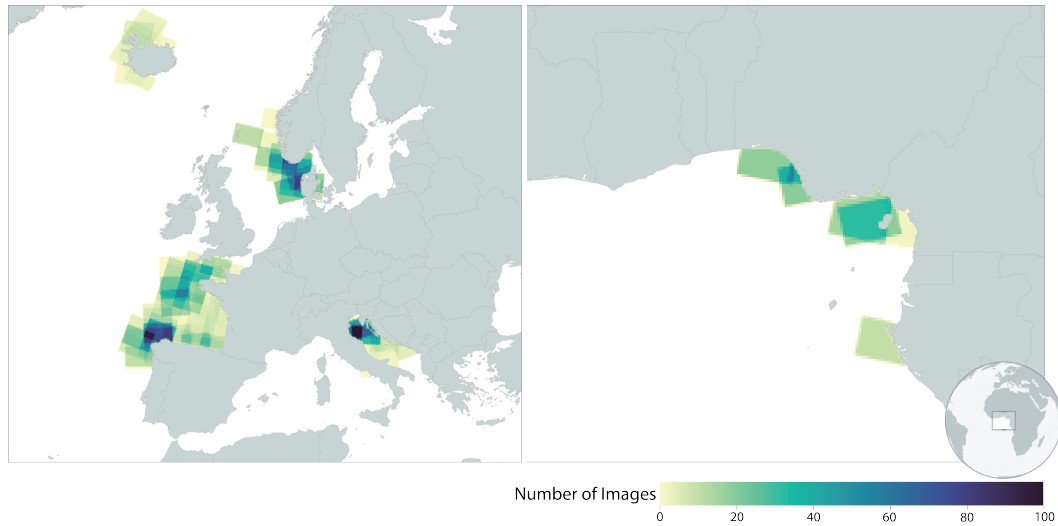

Figure 4: Geographic distribution of Sentinel-1 images used in the xView3 dataset. (left) European waters present a diversity of maritime objects, such as fishing and shipping vessels of all sizes, and offshore infrastructure. (right) The west coast of Africa is known to have substantial IUU activity.

a global terrain model on a 15-arc-second interval grid that we clipped to match the extent of each SAR image. All ancillary data—bathymetry, wind speed and direction, wind quality, and land/ice masks—are reprojected to UTM and resampled to 500-meter pixel spacing. We selected only SAR images for which corresponding ancillary rasters were available.

**Step 4. Detect objects automatically**. Global Fishing Watch (GFW) maintains an extensive database of vessels and offshore infrastructure derived from Sentinel-1 imagery [44]. The detection approach combines a well-established CFAR ship detection algorithm [34, 41] with a ConvNet classification and regression model to identify (and filter out) noise and estimate length. For xView3-SAR, we selected over 161,000 detections from 2020 spread over several geographic areas.

**Step 5. Correlate AIS to automated detections**. GFW also maintains an extensive database of processed AIS data [37]. Matching AIS messages to the respective vessels in SAR images is challenging; the timestamps of these measurements do not coincide, and AIS messages can potentially match to multiple vessels appearing in the image, and vice versa. We used a probabilistic model that determines {AIS, SAR-detection} pairs based on AIS records before and after the time of the image and the probability of matching to any of the vessels in that image [14]. This method performs significantly better than conventional approaches such as interpolation based on speed and course. This step is crucial in providing labeled data for ML models, as well as for validating external labeling protocols. Application of this approach to AIS-SAR correlation for labeling a ML dataset represents a novel aspect of our work (for details of the matching procedure we refer to [14] )[7].

**Step 6. Classify AIS to characterize vessels**. We supplemented matched detections with GFW's estimates of vessel type and activity (i.e. fishing vs. non-fishing classification). These identities were determined by combining information from available vessel registries with predictions from a ConvNet [15] that learns vessel movement patterns to estimate vessel type and activity.

**Step 7. Detect objects manually**. We trained professional labelers to visually identify the bounds and the category of objects in 437 SAR images. These manual annotations provided approximately 176,000 labeled detections, along with the annotator's confidence in identifying the objects. Our instructions to labelers can be found in Appendix G. Sample annotations are shown in Figure 5.

---

[7]The matching algorithm can be accessed at the GFW's GitHub repository.

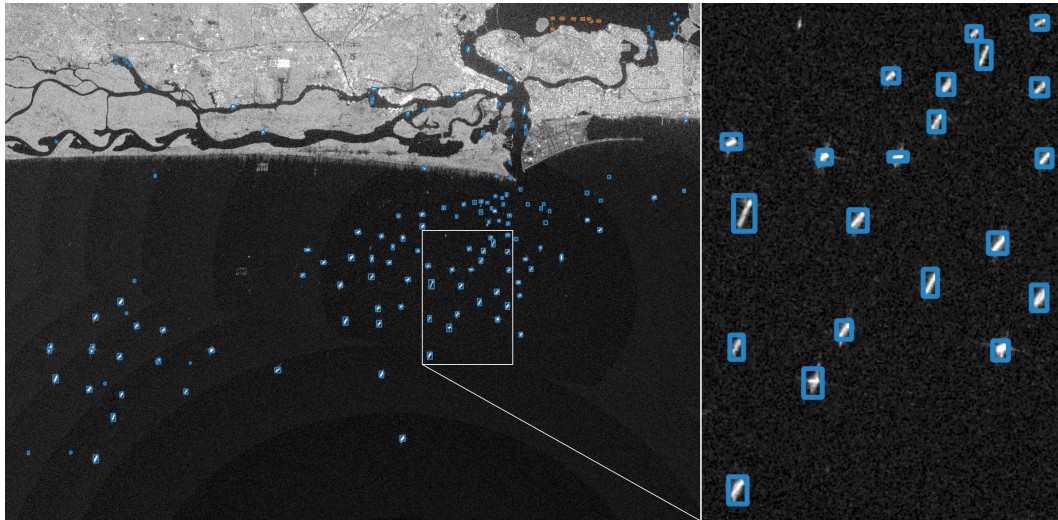

Figure 5: A Sentinel-1 SAR view near Lagos, Nigeria, showing bounding boxes of detected vessels (blue) and fixed infrastructure (orange, top right corner on left panel). Note the characteristic SAR speckle noise in the background (zoom in, right panel).

**Step 8. Combine automated and manual annotations**. Our labeling approach brings together AIS information and expert analysts to provide a comprehensive labeling system. Many instances of hand labels were also labeled by GFW's automated approach, while some instances were unique new labels (due to, for example, limitations of the current GFW algorithm to detect vessels close to shore). We note that an all-automated annotation approach would miss, among other things, vessels that do not broadcast AIS. On the other hand, an all-manual approach is labor intensive and costly, and introduces some degree of subjectivity as certain objects such as vessels, rocks, and offshore structures may show a similar visual signature in medium-resolution SAR. Furthermore, it is often the case that a fishing vessel cannot be distinguished from a non-fishing vessel by visual inspection. For these reasons, we combine automated and manual labels when both are available, and provide single-source labels otherwise (Figure 6). By correlating AIS information with information provided by human annotators, we are able to assign confidence levels—high, medium, and low—to each label (Appendix A). Overall, we generated 243,018 labels, with 39.1% having both automated and manual annotations, 33.4% manual only, and 27.5% automated only (Appendix D).[8]

**Step 9. Partition the data**. For the xView3 Computer Vision Challenge, we partitioned the data into four sets: *train*, 554 images; *validation*, 50 images; *public*, 150 images; *holdout*, 237 images. The *train* set contains only automated GFW labels, while all other sets contain labels created by combining automated and manual annotations (see Tables 2 and 3 in Appendix D for the breakdowns of the geographical and automated/manual annotation distribution for each data partition). The *train* and *validation* sets were provided to competitors for training and evaluating their ML models; the *public* set was used for the in-challenge public leaderboard; the *holdout* set was retained for final model performance assessment. We release all data but the *holdout* set.

## 4 The xView3 Computer Vision Challenge

The xView3-SAR dataset was developed for the xView3 Computer Vision Challenge, an international competition to detect and characterize dark vessels using computer vision and satellite SAR imagery. The competition, led by the U.S. Defense Innovation Unit (DIU) and GFW, launched on August 2021. Over 2,000 competitors used xView3-SAR to develop state-of-the-art maritime object detection and characterization algorithms. The Challenge aimed to deploy selected algorithms to support real-world practitioners in the fight against IUU fishing. Below, we summarize the ML tasks in the xView3 Challenge that used the xView3-SAR dataset. Although a detailed analysis of the winning models

---

[8]Nearly all automated-only labels are in the training set that contains no manual labels.

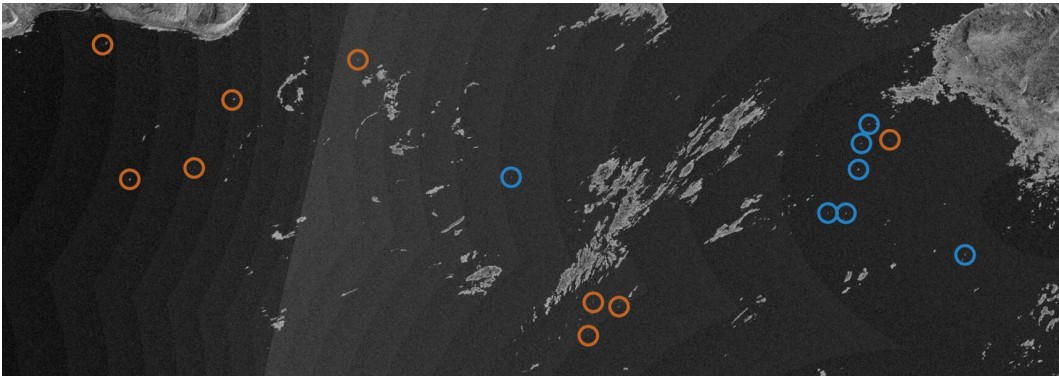

Figure 6: Mixed annotations along a complex shoreline in northwest Iceland. Blue circles are high confidence human annotations correlated with AIS data; orange circles are AIS-only annotations. This is a challenging scenario for vessel detection as many rocks can be confused with real objects.

from the challenge is beyond the scope of this paper, we offer some observations that users of xView3-SAR may find useful in section 4.2.

## 4.1 Summary of xView3 Challenge Machine Learning Tasks

**Maritime Object Detection**. To evaluate detection performance, the challenge considered "ground truth positives" to be human-made maritime objects in an image that were either (i) identified by a human labeler with high or medium confidence and/or (ii) identified via a high confidence correlation with AIS. Difficult aspects of this task included: (1) detailed shoreline delineation; (2) handling SAR artifacts such as ambiguities and sea clutter; (3) maintaining high performance across sea states, SAR acquisition configurations, and geographic domains. The challenge used the F1 score, $F1_D$, on all objects to measure performance on this task.

**Close-to-Shore Object Detection**. This task is of particular interest because there is a higher density of vessels closer to shore, and they can be difficult to differentiate from small islands or shoreline features. Radar signals can also interact with the surrounding land and structures. And other human-made objects such as piers or bridges are common. The challenge used an F1 score, $F1_S$, that compared predictions within two kilometers from shore (the 0-meter contour on the coregistered bathymetry map) to ground truth detections in those areas.

**Vessel Classification**. This task determines whether a maritime object is a "vessel" or "fixed infrastructure", such as offshore wind turbines, oil platforms, or fish farms. The challenge considered "ground truth positives" to be vessels in an image that were either (i) identified by a human labeler with high or medium confidence and/or (ii) identified via correlation with AIS. A standard F1 score, $F1_V$, over detections for which vessel information was available was used to measure performance.

**Fishing Classification**. Solvers were asked to further break down the "vessel" class into "fishing" and "non-fishing" classes. The challenge used a standard F1 score, $F1_F$, to measure performance, with positives defined as vessels in the "fishing" class. Ground truth labels for this task came from both (i) reported AIS data and (ii) the AIS analysis algorithm of [15], which shows 99% accuracy at classifying a vessel as "fishing" or "non-fishing" based on AIS information. Fishing classification annotations represent a novel contribution of xView3-SAR.

**Vessel Length Estimation**. Vessel size is key to discriminating IUU fishing activity as most dark fishing vessels tend to be small compared to cargo and passenger vessels that are likely to broadcast their AIS. The Challenge defined performance on this regression task using an "aggregate percent

error," $PE_L$:

$$PE_L = 1 - \min\left(\frac{1}{N}\sum_{n=1}^{N}\frac{|\min(\hat{\ell}_n, \ell_{\max}) - \min(\ell_n, \ell_{\max})|}{\min(\hat{\ell}_n, \ell_{\max})}, 1\right),\tag{1}$$

where $\ell$ is the true length and $\hat{\ell}$ the predicted length; $\ell_{\max}$ is the maximum length of a predicted or ground truth object.[9] Note that the majority of the vessels in the dataset are below 40 meters in length, meaning that their footprint on Sentinel-1 SAR may be relatively small (Figure 7)

**Overall Ranking Metrics**. The Challenge combined individual performance metrics to compute an aggregate multitask metric $M_R$ to rank submissions. $M_R$ was designed such that: (i) scores should range between zero and one; (ii) poor overall object detection should result in poor overall performance; (iii) advances on any of the other tasks should result in equal levels of score improvement:

$$M_R = F_D \times \frac{1 + F1_S + F1_V + F1_F + PE_L}{5}.\tag{2}$$

For the purpose of computing the ranking metric, we employed labels that were medium and high quality only as ground truth.

**Evaluation Constraints**. We aimed to deploy the winning challenge models to aid real-world anti-IUU efforts. Since anti-IUU practitioners often operate under limited computing resources and time is critical, a useful model must be computationally efficient. Competitors were asked to develop models that can run inference on any full SAR image, of about 29,400-by-24,400 pixels, in under 15 minutes on a computer with one Tesla V100 GPU, 60 GB RAM, and a server-grade CPU. As an example, while the first-place model required more than 63 hours of training time, inference for an entire SAR image requires around 13 minutes, allowing for near-real time vessel detection in production.

**Reference Model**. To introduce the xView3-SAR dataset and associated ML tasks, a reference detection and classification model was provided. This model is based on the Faster-RCNN architecture [32] and is intended to be a starting point upon which users can build rather than a well-performing baseline (Appendix B). The reference model code is available on the xView3 GitHub repository to enable the community to effectively use the dataset. The code for implementing the above metrics is also provided.

### 4.2 Summary of xView3 Challenge Results

The xView3 Challenge ran from August to November 2021. During this period, competitors submitted their predictions on the *public* set. At the conclusion of the challenge, competitors' models were evaluated against the *holdout* set (Table 1). Competitors adopted creative ML strategies to tackle the object detection, classification, and regression tasks. We provide a high-level overview of the best performing strategies, leaving detailed analyses to future researchers. We also provide a brief characterization of the winning model performance in Appendix C. We note that the DIU GitHub repository contains the code and detailed reports for each of the winning solutions.

The top solutions generally adopted a single-stage object detection strategy tailored for predicting small and tightly packed objects. Single-stage detection models allow for efficient inference to abide by the inference runtime limit. The first-place model used a pretrained CircleNet [51] for the encoder part and a U-Net [33] as the decoder to produce an intermediate feature map that is then operated upon by the model head. The head predicts an objectness map, offset length, and two dense classification labels—whether the object in question is a vessel and, if so, whether it is a fishing vessel. The regression head was modified to predict only the object length (in pixels).

All top solutions addressed data augmentation specific to the SAR domain. This is significantly more challenging than for RGB images because only a subset of existing image augmentations can be directly applied to SAR. For example, competitors extended the Albumentations library [4] to include custom-made augmentations for SAR images: random brightness and contrast changes, as well as various noise additions to mimic different speckle patterns. Competitors also explored strategies to

---

[9]$\ell_{\max}$ is set to 500 m to bound the maximum error, chosen based on the largest known vessel: Seawise Giant.

| Rank | Competitor | Aggregate | $F1_D$ | $F1_S$ | $F1_V$ | $F1_F$ | $PE_L$ |
|---|---|---|---|---|---|---|---|
| 1 | BloodAxe | 0.6177 | 0.7702 | 0.5310 | 0.9392 | 0.8425 | 0.6971 |
| 2 | selim_sef | 0.6047 | 0.7629 | 0.4768 | 0.9346 | 0.8079 | 0.7437 |
| 3 | Tumen | 0.5805 | 0.7395 | 0.4205 | 0.9479 | 0.8279 | 0.7289 |
| 4 | Skylight at AI2 | 0.5777 | 0.7322 | 0.4674 | 0.9293 | 0.8232 | 0.7253 |
| 5 | Kohei | 0.5717 | 0.7342 | 0.4527 | 0.9380 | 0.7910 | 0.7112 |
| | xView3 reference model | 0.1904 | 0.4302 | 0.1293 | 0.6891 | 0.3946 | — |

Table 1: Results of the xView3 Challenge on the *holdout* data partition. The xView3 reference model is a simple Faster-RCNN that naively estimates length (as a mere example) leading to a large percentage error, thus omitted from the table.

reduce artifacts inherent to the SAR image formation process, such as apodization in discrete Fourier space to remove diffraction spikes near strong reflectors (Figure 1, top row).

All models struggled to identify vessels near the shoreline (Table 1, $F1_S$). Since shorelines are often rocky and uneven, SAR image artifacts derived from multi-path effects and scattering, to name a few, are more common in these regions. No competitor developed specific methods to account for the complexity of detecting near the shoreline.

The combination of novel single-stage architectures, imbalanced class training, and data augmentation techniques, among others, resulted in substantial performance and efficiency improvements. Detailed documentations for the top five solutions can be found on the xView3 GitHub repository (Appendix C). The final competition leaderboard and resources are available on the xView3 Challenge website. The code and weights for the top five submissions are released as open source code.

## 5  Conclusion & Future Work

Illegal, unreported, and unregulated (IUU) fishing is an urgent problem causing immense harm to the marine environment. Locating IUU fishing and understanding its nature is a challenging task that is exacerbated by limited enforcement resources. Technology to efficiently and accurately identify vessels engaged in IUU fishing is therefore critical to the long-term health of global fisheries.

Synthetic aperture radar (SAR) imagery and automated machine learning (ML) can serve as a powerful tool for finding and characterizing vessels engaged in IUU fishing. Automating the detection and characterization of dark vessels in millions of square kilometers of imagery covering the world's ocean, and making the data and code publicly available, will enable governments and agencies to transform the way we manage our ocean.

We have constructed and released xView3-SAR, a maritime object detection and characterization dataset, covering 43.2 million square kilometers in 991 SAR images averaging 29,400-by-24,400 pixels each. We combined Automatic Identification System (AIS) and human-expert annotation to provide labels for ship detection, classification, and regression tasks. The xView3-SAR dataset is the largest of its kind by an order of magnitude, containing dual-band SAR images (VH and VV) with co-located ancillary rasters providing environmental context. The xView3 Computer Vision Challenge, based on xView3-SAR, raised awareness of the problem of IUU fishing in the larger ML community, promoting the development of ML models that perform detection, classification, and length estimation effectively, outperforming current standard methods.

Toolchains and utilities based on xView3-SAR and the xView3 Challenge have already been deployed to support both government and non-governmental organizations in their ongoing fight against IUU fishing, highlighting the need in bridging the fields of ML and remote sensing.

Advances in multi-scale representation learning, contextually aware models, resource-efficient computer vision, positive-unlabeled learning, and the use of auxiliary modalities, represent compelling areas for future work. By demonstrating the effectiveness of ML on large quantities of SAR imagery, and by providing the xView3-SAR dataset alongside the winning models from the xView3 Challenge, we hope to spur research that leverages the unique capabilities of SAR imaging technology to combat IUU fishing.

## Acknowledgments

This work is made possible by funding from the Department of Defense and our partners at the US Coast Guard, the National Maritime Intelligence-Integration Office, the National Oceanic and Atmospheric Administration, and Oceankind. We thank CDR Michael Nordhausen who liasoned relationships between subject matter experts in this domain. We thank the team at the US Coast Guard's Maritime Intelligence Fusion Center-Pacific for providing their decades-long expertise in IUU fishing, and John Mittleman for his continuing support and valuable guidance over the course of the challenge. Paul Woods at Global Fishing Watch provided invaluable insights into the use of AIS and IUU fishing.

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
