# OpenReview forum: "xView3-SAR: Detecting Dark Fishing Activity Using Synthetic Aperture Radar Imagery"
_NeurIPS.cc/2022/Track/Datasets_and_Benchmarks — NeurIPS 2022 Datasets and Benchmarks _

### Official Review · Reviewer_soKX · 2022-07-11
**Large-scale SAR dataset to fight illegal fishing**

**Rating:** 6
**Confidence:** 4

**Strengths:**

[S1] The dataset is analysis-ready Sentinel-1 data for 43 million square kilometers, with the correspondent bathymetry, wind speed and direction, wind quality, and land/ice masks. Nearly 250K objects are annotated using three approaches (automated and manual, automated, and manual only), and the confidence of the annotation is provided.
[S2] Five different tasks are introduced, a reference model is provided, and the submissions of the xView3 challenge are summarized/discussed. Also, some submissions (including the top ones) are available through the dataset website.
[S3] The models developed using this dataset may aid anti-illegal, unreported, and unregulated fishing efforts.


**Weaknesses:**

[W1] Limited geographic coverage. The dataset includes areas in Europe and West Africa. Even though the situation in the coast of West Africa is critical in regard to IUU fishing, regions like the Arafura Sea and the West Bering Sea are not covered.
[W2] Ground truth. The authors state that their labeling approach is unique, as they bring together both AIS information and expert analysts. However, combining this information seems to be a “common” practice (please, refer to “Relation to Prior Work”). Also, the amount of labels that do not match between the automatic and the manual process is high. Thus, I wonder how many objects are not detected or misclassified. Were there objects coming from AIS not detected?


**Additional Feedback:**

[1] Page 7, “The Challenge used standard a standard F1 score”. Maybe remove one standard?
[2] Page 8, “modified to predict only only the object” → “modified to predict only the object”
[3] Are you planning to release the holdout dataset?


**Clarity:**

The presentation is clear. The paper is well written. The dataset is well motivated.

**Correctness:**

The claims are correct. The data set construction is sound and well motivated. The data set and the code (including submissions to the associated challenge) are available.

**Documentation:**

OK. The paper provides meaningful information, and the website and repository provide lots of additional information.

**Relation To Prior Work:**

Some relevant works are not discussed in the paper. Particularly, previous SAR data sets for ship detection are not discussed. For example:
[1] B. Li, B. Liu, L. Huang, W. Guo, Z. Zhang, and W. Yu, “OpenSARShip 2.0: A large-volume dataset for deeper interpretation of ship targets in Sentinel-1 imagery,” in 2017 SAR in Big Data Era: Models, Methods and Applications (BIGSARDATA). Beijing: IEEE, Nov. 2017, pp. 1–5.
[2] S. Xian, W. Zhirui, S. Yuanrui, D. Wenhui, Z. Yue, and F. Kun, “Air-sarship–1.0: High resolution sar ship detection dataset,” J. Radars, vol. 8, no. 6, pp. 852–862, 2019.
[3] H. Xiyue, W. Ao, Q. Song, J. Lai, H. Wang, and F. Xu, “Fusar-ship: a high-resolution sar-ais matchup dataset of gaofen-3 for ship detection and recognition,” SCIENCE CHINA Information Sciences, 2020.
[4] Airbus Ship Detection Challenge: https://www.kaggle.com/c/airbus-ship-detection


**Summary And Contributions:**

The xView3-SAR dataset provides 991 Sentinel-1 images (29,400 x 22,400 pixels, on average), preprocessed using Sentinel-1 Toolbox (orbit correction, noise artifacts removal, radiometric calibration, terrain correction, and reprojection). Note that only the VH and VV bands are included (10 m/px). Also, bathymetry, wind speed and direction, wind quality, and land/ice masks are provided (all reprojected to 500 m/px). The dataset includes nearly 250K annotated objects, coming from automated and manual annotations (39%), automated processing (28%), and manual only (33%).
Even though this is not the only dataset of its kind, very few analysis-ready SAR datasets are available. Moreover, the area covered by this dataset and the number of annotated objects is significantly greater than its competitors. Furthermore, the paper proposes five tasks, and a nice summary of the solutions posted in the xView3 challenge.

---

> ### Author Response · Authors · 2022-08-24
> **Response to Reviewer 6**
>
> > Limited geographic coverage.
>
> This is addressed in the main response in “geographic coverage.”  We agree with the concern, and have attempted to communicate the limitations of the dataset appropriately.
>
> > is the labeling approach novel? How many objects were not detected or otherwise misclassified?
>
> We appreciate this question, and we have addressed this point above in the “labeling procedure” and “matching system” sections. While we agree that AIS data has conceptually been used alongside SAR before, the sophistication of our matching system, the size of the dataset, and the multitude of tasks for which we provide labels (including the unique “fishing classification” task that is performed purely from the SAR image) represent distinguishing characteristics of this work.
>
> We believe that there is a misconception regarding a lack of corroboration between automatic and manual labeling processes stemming from the following sentence in our paper: "Overall, we generated 243,018 labels, with 39.1% having both automated and manual annotations, 27.5% automated only, and 33.4% manual only."
>
> However, the above numbers include the training set which contains AIS labels only. A more detailed breakdown is in Table 3 in the appendix. A large fraction of labels that were identified by automated AIS methods were further confirmed by human annotation. There was an additional percentage of instances picked up by human annotators only. This makes sense because the automated labels are going to miss, among other things, "dark vessels" and those objects close to shore (as described in the “labeling procedure” and “matching system” sections above).
>
> We will edit the last sentence in Step 8 to read: "Overall, we generated 243,018 labels, with 39.1% having both automated and manual annotations, 33.4% manual only, and 27.5% automated only. Note that approximately 96 percent of the automated only labels are in the training set which contains no manual labels." We will additionally add the footnote: "Please see Table 3 in the Appendix for a detailed breakdown of label source by data partition."
>
> Finally, there is a fundamental limitation with imagery resolution which can result in objects not being detected. At 20m per pixel, GFW’s AIS database may contain vessels that are under 10m in length which are then not corroborated by humans.
>
> > Some relevant works are not in the Related Works section.
>
> We appreciate the suggestions, we have added these to our related work section.
>
> > Typos on pages 7 and 8.
>
> Corrected.
>
> > Are you planning to release the holdout dataset?
>
> We have already released the holdout dataset and it is available for download on our dataset website [1].
>
> [1] https://iuu.xview.us/

---

### Official Review · Reviewer_JVyd · 2022-07-18
**A paper with a significant contribution towards detecting illicit fishing activities**

**Rating:** 6
**Confidence:** 4
**Correctness:** The claims made in the paper seams co…

**Strengths:**

The paper presents a data set for monitoring illegal or “dark” fishing activity, which is an important task that may have an impact on ocean life and sustainability. There are currently no available data sets for this task (that I found) that combines both AIS and SAR images. The data set is also mostly manually labeled and consists of more than 200.000 labels. The contribution of the paper is therefore significant.

**Weaknesses:**

The data set only cover areas in Europe and west Africa. Models trained on it may not perform well in other parts of the world, such as Asia.

The relationship between the ML tasks in Section 4 and detecting IUU fishing activities should be made much clearer. In particular, an explanation of how predicted fishing vessels are identified as IUU (or not IUU) is missing.

**Additional Feedback:**

The paper could have a figure explaining how the ML tasks in Section 4 and detecting IUU fishing is related. The paper should be revised as it stands now, according to my comment below.

My current score assumes that the authors will publish the paper without updating it any further.

**Clarity:**

The clarity for most parts of the paper is ok. See my comment under "weaknesses".

**Documentation:**

The data set is well documented. The NeurIPS paper checklist is located inside the supplementary materials.

**Ethics:**

I cannot see any ethical problems with the data set or the paper.

**Relation To Prior Work:**

The relation the prior work is sufficiently described.

**Summary And Contributions:**

The paper addresses unsustainable fishing practices by introducing a data set for detecting dark vessels that attempt to evade monitoring systems. The paper also present the xView 3 Computer Vision challenge, which is a competition to detect dark vessels.

---

> ### Author Response · Authors · 2022-08-24
> **Response to Reviewer 5**
>
> > The dataset only covers area in Europe and West Africa. Models trained on xView3-SAR may not generalize.
>
> This is a valid concern, and we agree that users should test the models on the specific area of interest to determine region-specific performance numbers if they aim to use models developed using xView3-SAR in other areas.  This is addressed in the main response in “geographic coverage.”
>
> > The relationship between the ML tasks in Section 4 and detecting IUU fishing activities should be made much clearer. In particular, an explanation of how predicted fishing vessels are identified as IUU (or not IUU) is missing.
>
> This is addressed in the main response in the “dark vessels and illegal fishing” section.

---

### Official Review · Reviewer_eWZM · 2022-07-22
**A promising large-scale dataset for SAR based ship detection**

**Rating:** 7
**Confidence:** 4
**Correctness:** Yes.
**Clarity:** Yes.

**Strengths:**

1. It is a first large-scale ship detection dataset with SAR images and curated labels.
2. There's rich label information: labels include bounding box and ship types, and the label source and confidence level are also provided.
3. A successful challange has been completed with clearly defined taks and promising results.
4. The data and codes are convenient to access.
5. The paper is well written.


**Weaknesses:**

1. Main concern is the accuracy and confidence of the labels, as the objects are very small in a big scene and SAR images are noisy.
  - How confident is GFW? There are labels which have no human annotation, what are the confidence levels with GFW only labels?
  - How confident are labelers? How many labelers are there? Are there repeated labeling for each object and the final label is major voted?
  - The distribution of different confidence levels is not provided, e.g., how many objects are "low", how many objects are "high".
2. As mentioned in the paper, it is not global.
3. The definition of "dark vessel" and its role in this paper is not very clear. The dataset seems to have no information about whether a vessel is a dark vessel.
4. In related works some previous ship detection datasets are cited but no straighforward comparsion. Maybe more specific comparison (e.g. a table of the important characteristics of a dataset like number of ships, resolution, label info, etc) would be helpful.

**Additional Feedback:**

Some minor points:
- In introduction, maybe add a citation to "MS-COCO".
- Typo? In Section 3 step 9, "The train set contains **only** automated GFW labels **only** whereas all other sets...."

**Documentation:**

Yes.

**Ethics:**

No.

**Relation To Prior Work:**

Yes.

**Summary And Contributions:**

This paper introduces a large-scale dataset for SAR based vessel detection, and presents an overview of the tasks, baseline model, winning solutions in the corresponding computer vision challenge.

---

> ### Author Response · Authors · 2022-08-24
> **Response to Reviewer 4**
>
> > Main concern is the accuracy and confidence of the labels, as the objects are very small in a big scene and SAR images are noisy.
>
> > How confident is GFW? There are labels which have no human annotation, what are the confidence levels with GFW only labels?
>
> > How confident are labelers? How many labelers are there? Are there repeated labeling for each object and the final label is major voted?
>
> We address these concerns in the main response in the “Labeling procedure” and “Matching system” sections.
>
> The confidence levels employed by human annotators and their definitions are in Appendix F and replicated here for convenience:
>
> * High confidence: I am absolutely sure it is a vessel.
> * Medium confidence: I am not so sure, but it looks like a vessel.
> * Low confidence: I am mostly guessing it is a vessel.
>
> Further, we asked the annotators to label an object as a non-vessel if and only if they are sure the object is a non-vessel.
>
> Overall, across object classes, 49.5 percent of the objects are of high confidence; 24.0 percent are of medium confidence; 26.4 percent are of low confidence.
>
> For the vessel class, 35.6 percent of the objects are of high confidence; 30.1 percent are medium confidence; 33.4 percent are low confidence.
> * The distribution of different confidence levels is not provided, e.g., how many objects are "low", how many objects are "high".
>
> Please note that the labels released as a part of the dataset employ a different set of confidence level definitions; the confidence levels and their definitions are stated in Appendix A.
>
> As for the distribution, 38.9 percent of the objects are of high confidence; 41.2 percent are of medium confidence; 19.9 percent are of low confidence. Among the vessel objects, 20.0 percent of the objects are of high confidence; 56.3 percent are of medium confidence; 23.7 percent are of low confidence. Among the non-vessel objects, 80.3 percent of the objects are of high confidence; 19.7 percent are of medium confidence.
>
> For the purpose of computing the ranking metric, we employ labels that are medium and high quality as ground truth. Low-confidence labels are not used in computing scores for the public leaderboard or the final challenge ranking.
>
> > The dataset is not global.
>
> This is addressed in the main response in the “geographic coverage” section.
>
> > The definition of "dark vessel" is not clear.
>
> This is addressed in the main response in the “dark vessels and illegal fishing” section.
>
> > The related works section could use more detail.
>
> We will add further details to the related works section in line with the comments you and R6 have provided. We will also add a table to the Appendix that offers a detailed breakdown of related datasets in this space.

---

> > ### Comment · Reviewer_eWZM · 2022-08-29
> > **Thanks for the response**
> >
> > Thanks for the response. Most of my concerns are addressed. I will keep my score and recommend acceptance with high confidence.

---

### Official Review · Reviewer_ghqF · 2022-07-25
**An interesting, thoroughly validated dataset**

**Rating:** 8
**Confidence:** 3
**Clarity:** The paper is well written.

**Strengths:**

1. Well defined (and implemented) metrics seem designed to ensure that the outputs of models trained against this dataset are useful to organizations aiming to combat illegal fishing. Therefore, strong performances on this dataset should correlate well to utility for those organizations.
2. The use of this dataset in a competition ensures its accesibility to ML practitioners. In addition, it provides practitioners with a range of well-implemented baselines upon which they can build.
3. From a machine learning perspective, this is an interesting and challenging multi-modal task

**Weaknesses:**

I had trouble finding the data processing code (steps 2 and 3 in section 3). Sharing this code would allow for individuals to extend this dataset, or to run it on novel geographies.

**Additional Feedback:**

None

**Correctness:**

The authors spend a considerable amount of time validating their dataset collection approach, both in terms of the data itself and the metrics used.

**Documentation:**

There is thorough documentation (in multiple sources). In addition, the existence of competition-winning code provides a base against which future research can be built.

**Ethics:**

None known

**Relation To Prior Work:**

Prior works in this area are cited

**Summary And Contributions:**

A dataset of Sentinel-1 SAR imagery with labelled vessels (and non-vessels, e.g. wind turbines). This dataset is globally distributed (although not globally representative), and consists of 991 Sentinel-1 images containing 243,019 verified maritime objects.

The goal of this dataset is to identify illegal "dark" fishing vessels, which have their Automatic Identification System turned off.

---

> ### Author Response · Authors · 2022-08-24
> **Response to Reviewer 3**
>
> > I had trouble finding the data processing code (steps 2 and 3 in section 3).
>
> We have provided an updated link to the data preprocessing code in the main response and updated the paper to point to the link [1] as well.
>
> [1] https://github.com/DIUx-xView/xview3-reference/tree/main/reference/data_processing

---

### Official Review · Reviewer_iwi2 · 2022-07-25
**Review for xView3-SAR manuscript**

**Rating:** 8
**Confidence:** 5
**Clarity:** The paper is well written, with only …

**Strengths:**

This submission has a number of strengths. First, the scale of the introduced dataset (1000 large-scale SAR images with 250,000 labels) ensures that it capture a wide distribution of maritime vessel signatures.

As the researchers themselves express, detecting maritime vessels is an extremely timely + relevant problem, as governments try to crack down on illegal fishing.

It is also very impressive the the authors have already used this dataset in hosting a large vessel-detection challenge. Per the specified details of this event in the manuscript, the challenge was well-designed and well-attended. That this dataset is already being used by the wider research community makes it a particularly strong submission.

**Weaknesses:**

As stated above, my main concerns revolve around 1) the amount of detailed provided that describes how the dataset was curated; 2) the framing of the paper as "detecting dark vessels"; and (3) the use of AIS in the labeling process.

I further detail these concerns in the "additional feedback section below".

**Additional Feedback:**

General comments are numbered below, and are presented in roughly chronological order.

1. It would be helpful to have different names for 1) the dataset, and 2) the ship detection contest you host using the dataset. This was confusing at first.

2. "EO" usually refers to "earth observation". For describing optical imagery, you can just say "optical".

3. Do not use grouped citations in Section 2. Clearly summarize the contributions of each referenced paper independently.

4. Page 3, there's a typo in "acquition modes".

5. What are the physics behind why vh/vv backscatter are better suited to distinguish vessels/sea clutter/oil slicks? Explaining why each of these bands perform well for different identification tasks would be very informative to the reader.

6. What causes the radar interference artifact in Figure 2, and where is this explained? Is this caused by the wind turbines? Or is it an artifact in image collection.

7. I see in the Supplement that you include information on the distribution of detected vessel lengths. I think you need to include an overview of this finding in the main text, as it is important to communicate to readers that the detected vessels and the effective resolution of the S1 satellite have similar length scales. (20m x 20m for S1, most vessels <40m length, presumably much smaller width).

8. Page 4, Step 2, first sentence: Imagery is a singular noun.

9. Step 3: How is reprojection done? Nearest neighbor? Why are S1 OCN files reprojected to WGS84 before projection into UTM coordinates? Also, I assume OCN product contain wind speed measurements? Be explicit.

10. Step 4: This is a big portion of your label collection process, and it would be helpful to explain somewhere why you don't just use CFAR for all your collected labels. If CFAR can be trusted, why go through the process of hiring enumerators? If it can't why do you use it here?

11. Step 5: Need a lot more information on how the probabilistic model described here matches AIS messages to detected ships. What if AIS messages aren't available for the prediction, as they will be ~20% of the time for dark vessels. What is there's no good match between AIS and the detected ship?

12. Step 6: Again, this step needs to be explained in much more detail. What happens when there is no AIS data available or GFW estimates for ships detected via CFAR?

13. Figure 5 -- where is this introduced or discussed?

14. You need much more emphasis on how this dataset pertains to "dark vessels". As far as I can tell, the proposed dataset is a more generic maritime ship detection dataset. This needs to either be clarified, or you need to reframe the paper away from the particular case of dark vessels.

15. Figure 6 and page 7, para 1: What is the AIS-only annotation? Is this the result of the automatic labeling process? If so, CFAR uses imagery so detection, so I don't believe it would qualify as AIS-only. Need to be more clear.

16. Table 1: Why is the reference model performance so low? Presumably now that the contest is completed, you should adopt the best performing models as your reference.

17. The results of the challenge are interesting, but I think they are best suited for somewhere else where you can go deeper into the details (blog post, article, etc.). Already, you're trying to communicate so much in this one manuscript, and I think you are better served by explaining the dataset curation process more comprehensively. Here, you could look into condensing section 4.1, removing section 4.2 (or just move it to the Supplement) and condensing/removing most of 4.3; you can then use this space for fuller descriptions of how you created the dataset, and in particular the different methods of label collection.

18. Reintroduce acronyms in the conclusion.

**Correctness:**

As far as I can tell, all claims in the submission are substantiated. The dataset is constructed in a sound way.

**Documentation:**

There needs to be more documentation on how exactly the labels were created for the xView3-SAR dataset.

Apart from this, the reviewer easily accessed and reviewed the dataset. A hosting, license, and maintenance plan is described.

**Ethics:**

Not applicable.

**Relation To Prior Work:**

The relation to prior work is discussed in the penultimate paragraph on page 3. This review seems to describe the main contribution in this space, but ideally the authors include more detail about similar resources.

Here, I envision a tabular overview of related maritime detection dataset, with columns representing: Number of images, num. labels, type of SAR data, processing applied, locations of imagery, time of collection, method of label collection, etc.

This should then go in the Supplement if there's no room in the main text.



**Summary And Contributions:**

This paper introduces the xView3-SAR dataset: A collection of analysis-ready SAR images on average 29,400 x 24,400 pixels each, paired with auxiliary raster data and 243,018 verified maritime object labels.

The dataset is introduced and explained, and then the authors describe the xView3-SAR ship detection competition that was completed using the eponymous dataset.

Overall, this is a very strong paper. I have some concerns about 1) the amount of detailed provided that describes how the dataset was curated; 2) the framing of the paper as "detecting dark vessels"; and (3) the use of AIS in the labeling process, given that ~20% of fishing vessels are estimated to be "dark". That being said, these concerns can be address the revision period, which would improve an already compelling submission.

Please find a full set of my comments below.

---

> ### Author Response · Authors · 2022-08-24
> **Response to Reviewer 2 (2/2)**
>
> > How does this pertain to "dark" vessels?
>
> The issue is addressed in the main response in the “dark vessels and illegal fishing” section.
>
> > What is AIS-only annotation in Figure 6 and paragraph 7? If CFAR was used, then any derived labels do not count as "AIS-only".
>
> This is a valid point.  We will clarify that “AIS-only” means “automated labeling using computational correlation between AIS signals and CFAR detections” as described in [1].
>
> > Why is the reference model performance low?
>
> We retain the “reference” model here as a comprehensive, easy to extend codebase versus a “baseline” for performance comparison purposes.  We have re-emphasized this in the text.  We agree that future studies would most likely want to use the winning xView3 model as a “baseline” for performance measurement. This is one of the anticipated outcomes of the challenge, and the reason we open-sourced the top five models.
>
> > You could condense the writeup...
>
> The reviewer makes a good point, thanks. We condensed sections 4.1 and 4.3, and merged section 4.2. We also expanded the text on the dataset (as described above).
>
> > Reintroduce acronyms in the conclusion.
>
> We have expanded the acronyms in the conclusion.
>
> [1] https://eartharxiv.org/repository/view/3239/

---

> ### Author Response · Authors · 2022-08-24
> **Response to Reviewer 2 (1/2)**
>
> > Different names for the dataset and prize challenge.
>
> We now make clear in the manuscript that xView3 is the Computer Vision Challenge and xView3-SAR is the dataset that came out of this contest and the main focus of this paper. We would like to keep the dataset name linked to the specific contest as there have been previous iterations of the DIU’s xView Challenge Series [1]. For instance, xView2 used optical imagery to perform automated building damage assessment (BDA) after natural disasters; this challenge produced the xBD dataset. However, the community has been confused by the difference in naming between the challenge and dataset since.
>
> > "EO" refers to "earth observation".
>
> We changed “EO” to “optical”.
>
> > Do not use grouped citations.
>
> Given that we cite nearly 30 articles in our related work, we respectfully submit that it is not feasible to summarize the contributions of each one within the space limits of the paper. We also note that the section on previous ML work is only a summary to contextualize the reader given that the focus of the paper is the dataset creation. We have gone back and attempted to ensure that our descriptions are as faithful to the original works as possible.
>
> Additionally, grouped citations are a common practice within the NeurIPS community. For instance, last year’s best papers in the Datasets and Benchmark track employed a similar grouped citation strategy [2], [3].
>
> > Typo on page 3.
>
> Corrected.
>
> > What are the physics behind what makes VV/VH better for vessel/oil/sea clutter detection?
>
> The co-polarization channels (HH and VV) are waves that are horizontally/vertically transmitted *and* received by the sensor.  In these co-polarizations,  small variations in surface texture can have varying reflectivity behaviors. The cross-polarization channels (VH and HV) represent the proportion of the signal that changes its polarization before it is received due to interacting with objects at the surface. Over relatively flat areas, such as the ocean surface, only a small fraction of the signal returns polarized. Thus, the VH and HV bands show a better separation between vessels (strong returns of polarized signal) and the sea clutter (weak polarized signal).
>
> > What causes the radar interference in Figure 2?
>
> Ground radio-frequency emissions in the band of the SAR frequency can cause bright features in the SAR image. The radar interference observed in Figure 2 could be from a weapons system as discussed in depth in a Bellingcat article linked here [4]. However, since many powerful radar emitters could cause interference with the SAR imagery collection process and we are not certain as to the exact cause, we explain that there is “radar interference” in the caption without being explicit as to its source (we clarified in the caption that this is a potential ground radio-frequency interference). This is not related to wind turbines, and it’s a rather common artifact in SAR scenes that adds to the challenge of object detection and characterization with SAR imagery.
>
> > Information about the distribution of vessel lengths is in the Supplement.
>
> We have moved some of this material up to the main text.
>
> > Typo on Page 4.
>
> Corrected.
>
> > How is reprojection done?
>
> For the SAR scenes that come in GeoTIFF format, we first extract the coordinates of the center pixel, and then find the corresponding UTM zone. We then reproject using the `gdalwarp` tool with bilinear interpolation. For the OCN rasters that come in NetCDF format unprojected, we convert these files to GeoTIFF with Ground Control Points (GCPs) in the metadata, then reproject using the `gdalwarp` tool that can automatically interpret these GCPs (we use nearest neighbor for raster masks), resample to match the WGS84 SAR grids, and reproject to UTM. Geospatial calculations are usually more accurate in native spherical coordinates (WGS84).
>
> > Why not use CFAR for everything?
>
> This is addressed in the main response in “Labeling procedure” and “Why machine learning” above; we will add similar material to the text.
>
> > How does AIS matching work?
>
> This is addressed in the main response in the “matching system” section; we will add similar material to the text.
>
> > What happens if there is no AIS data available?
>
> This issue is addressed in the main response in the “matching system,” “labeling procedure,” and “geographic coverage” sections; we will add similar material to the text.
>
> > Figure 5 is never referenced in the paper.
>
> Thanks for catching this. We now introduce Figure 5 with the respective text regarding the scale and density of the objects in question, contrasting the more standard object detection tasks in computer vision.
>
> [1] https://xview.us/
>
> [2] https://arxiv.org/abs/2112.01716
>
> [3] https://arxiv.org/abs/2012.04035
>
> [4] https://www.bellingcat.com/resources/2022/02/11/radar-interference-tracker-a-new-open-source-tool-to-locate-active-military-radar-systems/

---

### Official Review · Reviewer_hYZt · 2022-07-27
**Nice dataset with proven public challenges**

**Rating:** 7
**Confidence:** 3

**Strengths:**

- The paper targets a crucial, meaningful, yet unsolved topic: illegal fishing detection. This kind of dataset will potentially enable detecting illegal fishing activities remotely using AI. Thus, a significant amount of economic loss can be prevented.
- It is the largest open-source dataset of its type by order of magnitude. Specifically, they combine AIS data, a state-of-the-art AIS-SAR matching algorithm, and expert human-analyst verification to construct xView3-SAR. Besides that, the quality of the dataset is high: 991 analysis-ready SAR imagery with a large size of 29,400-by-24,400 pixels, 243,018 annotations from experts, and a set of reference code to expedite users’ development process.
- They already ran the xView3 international challenge with a decent number of participants, which speaks to the soundness and quality of the dataset.

**Weaknesses:**

- Labels. Although authors claim that their labeling process is quite unique, as they bring together both AIS information and expert analysts to provide the annotations. Among 243,018 labels, 39.1% were both automated and manual annotations, 27.5% were automated only, and 33.4% were manual only. I do not understand why not provide automated and manual annotations for all labels. If it is because labeling the data requires labor effort, why not provide 100% automated annotations and manual annotations as much as possible. Data and labels are very important to any of the ML algorithms. Table 4 from the supplement shows there are only around 1% AIS-only labels. Is it because low-confidence labels are being filtered or what? There should be some comparison examples to help the reader understand the correctness of both automated and manual annotations. I think part of that should be in the main paper instead of the supplement as it will help readers a lot to grasp the idea of the labeling process. In summary, the labeling process/distribution is not as sound as other parts of the paper.

- Number of images. Although 991 was claimed to be the largest SAR dataset of this kind. The amount of the data is quite small compared to other domains. I wish authors keep expanding the dataset to stimulate this research area. Only 15 mins of training time on V100 speaks for the small scale of the dataset compared to popular areas like image/face recognition. I understand that it is not easy to get this kind of image but 900 is indeed small for the current deep learning model to learn the generalization.

- Number of modalities seems small to me. Only SAR images and ancillary data are provided. However, I feel like there can be more metadata/intermediate data can be provided. For example, the radar data must have some FFT results or other results before the final images. More data/modalities will definitely help improve the algorithm's performance.



**Additional Feedback:**

Overall, I enjoy seeing the real-world problem- Illegal, unreported, and unregulated (IUU) fishing drawing research attention, and I think this dataset will pave the way for enabling detect illegal fishing activities remotely using AI. I really appreciate that the authors put effort into this important real-world problem. Some improvements can be made in terms of the labels, modalities, and baseline performance.

**Clarity:**

Overall the paper is well written and clear. But I feel like not sufficient contexts related to SAR were given. It is important for the non-expert readers to understand the idea in the main paper instead of looking up everything in other references. I would suggest add more SAR related background.

- Is 'acquition modes' typo? Not sure about that.

**Correctness:**

Overall the correctness is good. A few comments:


- The train-validate-public-holdout split is fixed. Did the authors consider shuffling them with different random seeds before making the final decision? Are the estimated results/scores from models very different when using different splits? Why choose this specific/fixed split? Multiple different splits can help avoid data bias.

- The baseline model seems very bad. I feel like the authors should have some designed choice about the baseline algorithm rather than a simple fast RCNN.

**Documentation:**

Documentation looks quite good to me.

**Ethics:**

If potential criminals get this kind of data and use it to escape Illegal, unreported, and unregulated (IUU) fishing, what will happen? Are there any measures to prevent this from happening? There should be a session to discuss this since the original intention of this paper is to prevent illegal activities. However, in many cases, criminals just learn from the cases and develop their strategies to fool the algorithm.

**Relation To Prior Work:**

I suggest having a table in related work, which helps clarify what kind of related work is done previously. There can be multiple columns comparing the different aspects of the dataset: number of images, resolution, number of modalities, task supported, etc.

**Summary And Contributions:**

xView3-SAR dataset targets a critical yet unsolved topic: illegal fishing. The authors present the largest labeled dataset for training ML models to detect and characterize vessels from SAR (Synthetic aperture radar). The dataset consists of 900+ analysis-ready SAR images from the Sentinel-1 mission. On average, 29,400 by 24,400 pixels each image. The images are annotated using a combination of automated
and manual analysis. Along with the dataset, five tasks/benchmarks, including Maritime Object Detection, Close-to-Shore Object Detection, Vessel Classification, Fishing Classification, and Vessel Length Estimation, are presented. They also provide an overview of the results from the xView3 Computer Vision Challenge, an international competition using xView3-SAR for ship detection and characterization at a large scale.

---

> ### Author Response · Authors · 2022-08-24
> **Response to Reviewer 1 (2/2)**
>
> > What if criminals get this data?
>
> The reviewer brings up a valid and thoughtful concern regarding the usage of the open nature of the data and models to counter IUU interdiction effort. This was a question we considered deeply before launching the challenge.  Indeed, informed by our NGO and government partners, we assess that there is relatively low risk to IUU interdiction operations were adversaries to get a hold of Sentinel-1 data and/or the xView3 models for the following reasons:
>
> 1. Sentinel-1 data provides revisits over the same area every 6-12 days. Due to the relative infrequency of the visits of the satellite constellation, the data is unlikely to be used to make real-time interdiction decisions. Likewise, IUU fishermen cannot be certain as to when their vessels will be illuminated by the satellite constellation. On the other hand, maritime security organizations have access to additional imagery at far shorter intervals, which allows for the proper usage of the technology as intended.
>
> 2. Obtaining Sentinel-1 data, as well as obtaining, setting up, and running the xView3 models requires a level of technical sophistication that is likely well beyond the abilities of most IUU fishermen. Since these fishermen cannot operate the models or share their results in a coordinated fashion,there is unlikely to be a significant risk to making these data or models available to the public.
>
> 3. IUU fishermen are unlikely to have access to financial or R&D resources to invest in stealth technology to defeat imaging radar or other forms of satellite imagery, leaving them well-suited for automated detection from space.

---

> ### Author Response · Authors · 2022-08-24
> **Response to Reviewer 1 (1/2)**
>
>
>
> > Why not provide 100% automated annotations and manual annotations?
>
> We address this in our main response in the “Labeling Procedure” section.
>
> > There are only around 1% AIS-only labels.
>
> Since we merge AIS and manual labels where a correlation is present, many of the AIS-located vessels are merged into the “AIS/Manual” section of labels in Table 4 of the supplement. The exact process is addressed in the main response in the “Labeling Procedure” section above.
>
> We have expanded the labeling section in the main text to further clarify this process.
>
> > The amount of data is quite small compared to the other domains.
>
> As described above in the “Size of dataset” response, the xView3-SAR dataset is ~4.7x larger than MS COCO, and ~2x larger than ImageNet 2012 in terms of pixels.  We have attempted to better communicate that each of the 991 images is so large that it would yield over 10,000 256 x 256 patches (from a single SAR scene).
>
> > Only 15 mins of training time on V100.
>
> The 15 minutes stated in the manuscript refers to “inference” time over a full SAR image (of size ~29,400 x 24,400 pixels), not training. The xView3 prize challenge requirement was that a trained model should require under 15 minutes to make predictions on a single SAR image. This requirement is needed for implementation in near-real-time detection and characterization systems that operate with large amounts of SAR imagery. In documentation provided by the first place xView3 prize challenge winner on their GitHub repo [1] we can see that their model required >63 hours of training.  We have clarified this in the text.
>
> > Only SAR images and ancillary data are provided.
>
> We believe that the introduction of multiple sources of ancillary data, such as bathymetry, wind speed, and land/ice masks, are meaningful additional sources of information. It is possible to further distribute single-look complex (SLC) products that include complex-valued phase and amplitude information on the slant range (or radar coordinates). However, the inclusion of these products would make the challenge unapproachable to many who do not work with complex-valued data. Also, this product is distributed as individual (slightly-overlapping) sub swaths that must be stitched together to form the GRD images (the type we supply), which adds significant complexity in the processing. Additionally, the increased size of the dataset (by a factor of ~3-4x) would make it extremely difficult to distribute and download.  Because we have released the Sentinel-1 image IDs along with our dataset, researchers will be able to download the SLC data and perform their own analysis on this more “raw” form of the data if they wish.
>
> > Fixed train/validate/public/holdout split.
>
> We designed the dataset split to ensure that each partition contained images from a similar distribution of geographic areas.  To mitigate the possibility that the specific test set we chose would cause unreliable performance measurement, we compared performance on the public test set and holdout set – in all cases, deviations in performance measured on the public test set and holdout test set were negligible.  We note that the public test set and holdout sets were disjoint, having no shared images.  Furthermore, it is important to remember that the validation, test, and holdout sets had both manual and AIS derived labels, while the training set had only AIS derived labels (see “Labeling procedure” above).  This was done to ensure that the performance measurements were computed on labels that were as high quality as possible.  For perspective, our holdout set is quite large, containing enough pixels to create over 2 million 256 x 256 image patches.
>
> > Baseline model performance is low.
>
> We explicitly avoid referring to the reference model as a “baseline”. In Section 4.2, we mention that this model is meant to serve as a starting point upon which users successfully begin using the dataset, end-to-end. The reference model allows any user in the world to clone one repository and effortlessly begin experimentation. We will update our wording to emphasize that the reference model is not a baseline and is instead a comprehensive, easy to extend codebase, showing some strategies to ingest large SAR images into ML models.
>
> [1] https://github.com/DIUx-xView/xView3_first_place

---

### Author Response · Authors · 2022-08-24
**Main response to reviewers (1/7)**

We thank all reviewers for all the hard work and time put into their reviews. Your insights will help us ensure that our paper communicates our work in the clearest way possible. We are excited that the reviewers found our dataset and prize challenge to be timely and highly impactful. Your reviews demonstrated confidence in our sophisticated labeling methodology, in the size of our dataset in terms of amount of imagery data (>1400 gigapixels) and labels (>240,000), and our ability to validate the complexities of the dataset via an international prize competition

In response, we have attempted to address all reviewer comments in full. We have also expanded the manuscript to include additional information and further clarify the points raised by the reviewers. Below we provide a detailed response for the major concerns across reviewers, followed by a point-by-point response to each reviewer.

We grouped our response to common concerns into one “main” response. For ease of readability, we split each concern into its own OpenReview comment below:

---

> ### Author Response · Authors · 2022-08-24
> **Main response to reviewers (7/7)**
>
> **Why dark vessels and illegal fishing? [R2, R4, R5].**  We expanded the introduction of the dataset and ML problem to highlight the “dark vessels” problem as a key maritime application. By definition, dark vessels do not broadcast their location, this means that we do not have AIS information about these targets. Illegal, unreported, and unregulated (IUU) fishing vessels tend to be “dark vessels” because they do not wish to be detected by law enforcement or maritime security operations.  SAR imaging is one way to detect these dark vessels regardless of whether they transmit AIS. We correlated AIS data from GFW with SAR detections from our CFAR algorithm (see “Matching system” above), and manually annotated detections that did not correlate with any AIS messages (see “Labeling procedure” above). The latter set of annotated data – i.e., those with only manual labels – includes potential dark vessels that we have labeled. Furthermore, the end-to-end procedure we describe in this manuscript, from the data processing and matching to AIS to the ML detection and characterization approach (e.g. classifying fishing vs. non-fishing) is ideal for dark vessel detection and characterization at a sufficiently large scale to support, for instance, anti-IUU patrol planning for regional security organizations.

---

> ### Author Response · Authors · 2022-08-24
> **Main response to reviewers (6/7)**
>
> **Why machine learning? [R2, R6].** Conventional anomaly detection methods based on the Constant False Alarm Rate (CFAR) algorithm have been widely used to detect ships in SAR images due to their simplicity and efficiency. There are two main reasons, however, to seek a more modern approach to the ship detection and characterization problem. First, the CFAR algorithm requires determining parameters a-priori that are image specific. For example, the cutoff threshold to detect an anomaly based on the mean and standard deviation of the radar backscatter, which varies both across the image and from image to image. This optimal threshold is usually determined empirically for the dataset at hand through extensive experimentation, which hinders automated analysis at scale. Further, if new images from the same SAR sensor happen to display different statistical properties, which is not uncommon, a new threshold needs to be found. Machine learning approaches, in particular neural networks, can learn the statistical properties of the images and automatically determine what constitutes an anomaly.
>
> Second, any object characterization such as its length, type, orientation, and speed, must be performed as a secondary task after detection using other task-specific approaches. Neural networks, on the other hand, can perform regression and classification tasks jointly with object detection, allowing the user to retrieve all information about the objects on a SAR image at once and enabling a learned representation to be optimized for all of these tasks simultaneously. Moreover, it is possible to train a neural network to recognize (and disregard) land pixels on an image, removing the need for masking land masses a-priori or during post-processing.  For these reasons, ML approaches to ship detection and characterization on SAR imagery are highly desirable.

---

> ### Author Response · Authors · 2022-08-24
> **Main response to reviewers (5/7)**
>
> **Geographic coverage [R4, R5, R6].** In order to provide a consistent SAR training dataset for ML detection and characterization problems, two key aspects must be considered. First, we need areas with a sufficiently large number of images, with complete SAR coverage, and frequent SAR acquisitions so as to guarantee consistency in data quality and availability. Spatial coverage and temporal sampling of the Sentinel-1 satellites vary geographically depending on the mission’s objectives and priorities. For instance, European waters are fully covered and sampled more often than any other areas around the world; given that Sentinel-1 is maintained by the European Space Agency, this is perhaps unsurprising. Second, we need coincident AIS data of good quality and in sufficient quantity to be able to reliably match AIS to SAR detections (see Matching System above). AIS quality and availability varies globally. For instance, southeast Asia suffers from poor satellite AIS reception quality [1], recent restrictions to AIS availability [2], and it is a region known for having most of its fishing fleets not broadcasting their AIS [3]. In addition to these constraints, we need to keep the overall number of regions and images manageable in a dataset that many ML practitioners should ideally be able to work with. Thus, we selected our geographic locations so as to balance data availability and quality with diversity of objects (e.g. vessels, offshore infrastructure) and activity (e.g. IUU fishing, shipping, recreation). We have re-emphasized this in the text, and now also provide the code for our SAR processing pipeline [4] as well as the matching system [5] so users can extend the dataset to other areas of interest.
>
> [1] https://windward.ai/blog/mind-the-ais-gap/
>
> [2] https://www.reedsmith.com/en/perspectives/2021/12/catch-me-if-you-can-new-chinese-privacy-law-causes-ais-disruption
>
> [3] https://www.science.org/doi/full/10.1126/sciadv.abb1197
>
> [4] https://github.com/DIUx-xView/xview3-reference/tree/main/reference/data_processing
>
> [5] https://github.com/GlobalFishingWatch/paper-longline-ais-sar-matching

---

> ### Author Response · Authors · 2022-08-24
> **Main response to reviewers (4/7)**
>
> **Matching system [R1, R2, R4, R6].** Global Fishing Watch (GFW) has developed a sophisticated matching system to pair AIS messages with SAR detections, and has dedicated a full publication describing the algorithm and its improvements from previous approaches, so we refer the more interested reader to [1] for the details.
>
> To facilitate discussion, we highlight here the key steps required in this procedure to provide the general reader with a better idea of the challenges. AIS data were obtained from commercial satellite providers, and were processed using Global Fishing Watch’s data pipeline. The identities with lengths of all AIS devices that operated near the SAR images in both space and time were obtained from Global Fishing Watch’s massive database [2]. Most AIS positions did not correspond to the exact time when the SAR images were taken. Hence, to determine the likelihood that a vessel broadcasting AIS corresponded to a specific SAR detection, GFW first developed probability rasters of where a vessel was likely to be minutes before or after an AIS position was recorded. GFW mined one year of global AIS data, including roughly 10 billion vessel positions, and computed these rasters for six different vessel classes, considering six different speeds and 36 time intervals, leading to 1296 different rasters. This probability raster approach could be seen as a utilization distribution [3] —for each vessel class, speed and time interval—where the space is relative to the position of the individual.
>
> For the vast majority of vessels there was an AIS message right before and after the scene, and thus two probability rasters. Two methods were used to combine these probability rasters to obtain information about the most likely location: “multiply and renormalize the rasters” when positions are >10 minutes apart from the image,  and “weight and average the rasters” when positions are closer. A matrix of scores of potential matches between SAR and AIS is then computed, and matches are greedily assigned and removed in an iterative procedure. A threshold for accepting SAR to AIS matches was determined empirically yielding three classes: “likely matches' , “potential matches”, and “unlikely matches”. This threshold approach performs significantly better than a metric based on the distance between the SAR detection and the most likely location of the vessel, where the likely location is based on extrapolating speed and course of the position closest in time to the image (as is used in most studies combining SAR and AIS).  Application of this approach to AIS-SAR correlation for labeling a machine learning dataset represents a novel aspect of our work.
>
> [1] https://eartharxiv.org/repository/view/3239/
>
> [2] https://www.science.org/doi/10.1126/science.aao5646
>
> [3] https://pubmed.ncbi.nlm.nih.gov/19694144/

---

> ### Author Response · Authors · 2022-08-24
> **Main response to reviewers (3/7)**
>
> **Labeling procedure [R1, R2, R4, R6].** There are two main challenges regarding labeling the xView3-SAR dataset: locating the objects in the SAR images and pairing them with known information (e.g. AIS). Locating and identifying small objects (a few bright pixels) on large SAR images (tens of thousands of pixels) is a challenging task, for both humans and algorithms. Matching these objects to AIS information adds additional complexity as (a) the timestamp of the AIS messages and the SAR image is different as these are acquired independently, (b) the location of objects from both AIS and SAR is not always accurate, and (c) on crowded scenes the same AIS message can potentially match to several objects, and vice versa. Therefore, a sophisticated matching algorithm is needed (more on this below).
>
> There are two main reasons why we don’t have both automated and manual labels for every object. First, the CFAR-based detection used for automated labeling is not applied within 1 km from shore to mitigate inaccuracies with the shoreline delineation. As human activity increases with proximity to the coast, automated CFAR detection potentially misses a significant number of objects. Second, AIS messages are only available from vessels that broadcast their AIS. As such, we do not have a way to automatically label objects for which there is no AIS (these are the “dark” vessels that are often associated with IUU fishing). Human annotators, on the other hand, can identify most objects appearing on the SAR images all the way to the shoreline (to the best of their knowledge). This is, however, a labor-intensive and costly task; and on the medium-resolution SAR we use here (about 20 m per pixel), visual inspection cannot always distinguish between, e.g. a vessel, a fixed offshore structure, and a rock, as they all appear equally bright with a geometry that is similar to the human eye on a given band. For these reasons, we combine automated and manual labels when both are available, and provide single-source labels otherwise. Based on this, we also established confidence levels for the labels, detailed in Appendix F “Instructions to expert annotators”. We note that the user can always choose what type of labels to use when training the algorithm.

---

> ### Author Response · Authors · 2022-08-24
> **Main response to reviewers (2/7)**
>
> **Size of dataset [R1].** Each of the 991 remote sensing images in the xView3-SAR dataset contains multiple bands of a full satellite scan, hundreds of millions of pixels in size. This is the standard way that space agencies distribute these products. The satellite SAR swath (or sub-swaths) usually covers several hundreds of kilometers on the ground (on the order of 250 x 250 km in the case of Sentinel-1 GRD products). These images, therefore, need to be divided into many small patches in order to be analyzed with standard ML algorithms. This is contrary to standard computer vision datasets that usually come in analysis-ready sizes, and more akin to work done in, for example, digital pathology. Standard computer vision datasets also tend to be composed of natural RGB images, with pixel values ranging from 0-255 (or 0-1) conveniently represented by the “uint8” data type (1 byte). SAR images, on the other hand, consist of multi-band rasters of “float32” data type for the ground-range detected (GRD) product (4 bytes), and “complex128” for the single-look complex (SLC) product (16 bytes). To provide a better idea of the size of SAR images and the xView3-SAR dataset, below are some direct comparisons to well-established computer vision datasets:
>
> * xView3-SAR: ~29,400 x ~24,400 pixels x 2 bands x 991 samples => ~1422 gigapixels
> * MS-COCO: 640 x 480 pixels x 3 bands x ~328,000 samples => ~302 gigapixels
> * ImageNet 2012: ~469 x ~387 pixels x 3 bands x ~1,300,000 samples => ~708 gigapixels
>
> We see that the xView3-SAR dataset is in fact ~4.7 times larger (in terms of pixel size) than MS COCO, and ~2 times larger than ImageNet 2012. On top of this, xView3-SAR also contains lower resolution ancillary rasters that we have not accounted for in the calculation above. These are 5 additional rasters of 1km per pixel co-located with each SAR scene, one associated with each of the 991 images. Indeed, the large size of SAR images is one of the major challenges for machine learning applications. We specifically had to reduce the original floating point precision in order to “facilitate” the download of the xView3-SAR dataset, which is still ~2 TB in its current form.  Note that because we provide the image IDs and preprocessing code, it would be possible for interested researchers to reconstitute the dataset with, for instance, full floating point precision.

---

### Meta-Review · Area_Chair_S5Aa · 2022-09-07

**Recommendation:** Accept
**Confidence:** 5

**Metareview:**

This dataset is a large dataset of >80M sq. km. of synthetic aperture radar satellite imagery with >220k instances of dark vessels for illegal fishing prevention. It has clear societal benefits, scientific value and novelty All the reviewers rate very positive for this dataset. The authors have carefully addressed reviewers' comments and revised the manuscript. Therefore, this dataset is suitable to be accepted.

---

### Decision · Program_Chairs · 2022-09-16

Accept